# Understanding prompt engineering may not require rethinking generalization

**Victor Akinwande[1], Yiding Jiang[1], Dylan Sam[1] & J. Zico Kolter[1,2]**
[1]Carnegie Mellon University, [2]Bosch Center for AI

## Abstract

Zero-shot learning in prompted vision-language models, the practice of crafting prompts to build classifiers without an explicit training process, has achieved impressive performance in many settings. This success presents a seemingly surprising observation: these methods suffer relatively little from overfitting, i.e., when a prompt is manually engineered to achieve low error on a given training set (thus rendering the method no longer actually zero-shot), the approach still performs well on held-out test data. In this paper, we show that we can explain such performance well via recourse to classical PAC-Bayes bounds. Specifically, we show that the *discrete* nature of prompts, combined with a PAC-Bayes prior given by a language model, results in generalization bounds that are *remarkably* tight by the standards of the literature: for instance, the generalization bound of an ImageNet classifier is often within a few percentage points of the true test error. We demonstrate empirically that this holds for existing handcrafted prompts and prompts generated through simple greedy search. Furthermore, the resulting bound is well-suited for model selection: the models with the best bound typically also have the best test performance. This work thus provides a possible justification for the widespread practice of "prompt engineering," even if it seems that such methods could potentially overfit the training data.

## 1 Introduction

Generalization bounds provide statistical guarantees on the average-case performance of a learning algorithm's output. However, in the case of deep learning models, there is still debate about how useful such bounds can be: Zhang et al. (2021) highlighted that classical approaches for deriving generalization bounds are insufficient for explaining the generalization ability of deep learning, spurring a flurry of new approaches for deriving tighter generalization bounds for deep neural networks (Bartlett et al., 2017; Dziugaite & Roy, 2017; Neyshabur et al., 2017b). In the recent literature on generalization bounds for deep learning, a large focus has been on developing *data-dependent bounds*, or bounds that consider both the data distribution and the hypothesis space. Some of the best data-dependent bounds arise from the PAC-Bayes framework (McAllester, 1999) and are derived by bounding the KL divergence between a prior over the hypothesis space and the posterior yielded by the learning algorithm. However, although PAC-Bayes bounds led to the first non-vacuous generalization bounds for deep learning (Dziugaite & Roy, 2017), they are still too loose to be practically useful (Jiang et al., 2019) in most realistic settings. In fact, as Lotfi et al. (2022) have recently argued, many PAC-Bayes bounds with data-dependent priors, while non-vacuous, can be best described as validation bounds — i.e., the use of data-dependent priors effectively leverages held-out data in a manner similar to cross-validation, which undermines their ability to *explain* generalization.

Notwithstanding the lack of a clear theoretical basis, modern machine learning models are moving towards increasingly large pretrained models (Kaplan et al., 2020; Dosovitskiy et al., 2020). One prevailing paradigm is to use pretrained foundation models such as CLIP (Radford et al., 2021) or ALIGN (Jia et al., 2021) as feature extractors and provide weak supervision for a downstream target task via *prompts*, which are text descriptions of the desired tasks that are often significantly easier to obtain compared to full model weights or even a generic linear classifier over the last layer. The versatility and performance of prompting pretrained models have led to the rise of *prompt engineering*, an emergent paradigm in machine learning where practitioners carefully design the task specification in text or even learn the prompts in a data-driven fashion (Lester et al., 2021). For example, to obtain a two-class image classifier, one would write two sentences that describe the classes (e.g., "This is

Table 1: Comparison with existing state-of-the-art generalization bounds for test error on different datasets. We report both data-independent and data-dependent bounds ($\star$ indicates data-dependent prior and $-$ indicates that the bounds are not available). Note that different works use different architectures and analytic tools so direct comparison can be more nuanced. Nonetheless, our bounds on prompt engineering are significantly tighter than the existing PAC-Bayes bounds in the literature, often within a few percent of the actual test error.

| Dataset | Zhou et al. (2019) | Dziugaite et al. (2021) | Lotfi et al. (2022) | PAC-Bayes (prompt) |
|---|---|---|---|---|
| CIFAR-10 | $-$ | $0.230^\star$ | $0.582\,/\,0.166^\star$ | **0.063** |
| CIFAR-100 | $-$ | $-$ | $0.946\,/\,0.444^\star$ | **0.266** |
| ImageNet | 0.965 | $-$ | $0.930\,/\,0.409^\star$ | **0.319** |

a dog" and "This is a cat"), and the two sentences are turned into text embeddings which can be used to classify image embeddings. Despite its empirical success, little is understood of how and why prompting these pretrained models work and, in particular, why the method seems to suffer little from overfitting: manually tuning or even greedily optimizing prompts on a given training set often performs nearly as well on the corresponding test set.

In this paper, we demonstrate that rather simple analysis tools capture this behavior surprisingly well (under some assumptions). In particular, we show that classical PAC-Bayes bounds (McAllester, 1999), when applied to the discrete hypothesis class defined by prompts (and specifically with a prior given by a large language model), are often *remarkably* tight, even for large domains: for example, we achieve a generalization bound of 32% error for a full ImageNet classifier, which is within *6%* of the actual test error. This represents a *vast* improvement over existing bounds for deep learning, where achieving any non-vacuous bound on domains like ImageNet typically requires a great deal of effort; see, for instance, Table 1 for a comparison with other approaches. Perhaps more interestingly, our bounds do not depend on the training data as the prior approaches do [1] but instead depend on the pretraining data of pretrained model (e.g., CLIP) through the image encoder.

To summarize, we find that, unlike conventional deep learning models, prompting pretrained models does not suffer from vacuous generalization bounds, and one can readily derive a strong theoretical guarantee for using prompts via well-studied techniques. Overall, these findings suggest that, despite a large amount of automatic or manual tuning, prompt engineering is a principled approach for using these pretrained models that do not suffer the same lack of theoretical grounding as conventional deep learning models. On the other hand, it does introduce its own set of considerations, which we will discuss in the experiments section and conclusion.

## 2 RELATED WORKS

**Prompt Engineering.** With the advent of large pretrained models, prompting developed as a different yet effective method to harness the abilities of these large models with limited labeled data (Brown et al., 2020; Le Scao & Rush, 2021; Liu et al., 2023). The flexibility of prompting has enabled a wide range of new capabilities unavailable to previous machine learning models, leading to a significant effort to document successful prompting methods (Bach et al., 2022) in both classification and text-to-image generation. One downside of prompting is that the performance varies greatly depending on how the prompt is phrased. To address this issue, several methods have been proposed to learn "optimal" prompts given labeled data, which empirically performs well and is parameter efficient (Lester et al., 2021; Li & Liang, 2021; Gao et al., 2021; Zhou et al., 2022a;b). A limitation of data-driven methods is their tendency to learn "soft" prompts or embedding vectors that do not correspond to specific tokens. Moreover, from a learning theoretic perspective, the continuous nature of soft prompts, combined with transformations by non-linear models, results in a complex hypothesis space, making it less amenable to theoretical analysis. In contrast, another line of work uses gradient-based methods to learn prompts that consist of discrete tokens that can be mapped to natural language (Wen et al., 2023). This work studies the theoretical guarantees of the latter

---

[1] Data-dependent priors primarily refer to the setting where a portion of training data is used to obtain a prior that is closer to the final posterior. We note that in our setting, the training of pretrained models uses data from different distributions and does not use any training data from the task of interest.

approach, that is, why these discrete prompting methods seem to work without any overfitting, and our analysis extends to the methods proposed in Wen et al. (2023).

Prompt engineering has been extended to computer vision through CLIP (Contrastive Language-Image Pretraining) (Radford et al., 2021). CLIP combines an image and language encoder trained jointly to minimize a contrastive loss, enabling it to perform classification tasks based on natural language instructions. Examples include object recognition, image caption generation (Tewel et al., 2021), and zero-shot image classification using textual descriptions even for unseen labels.

**Generalization bounds.** Generalization bounds are upper bounds on the test error of a model. Deriving such bounds for deep learning has been difficult, and most are usually vacuous (Zhang et al., 2021; Jiang et al., 2019; Dziugaite et al., 2020). Many well-studied tools in statistical learning theory are fundamentally limited when it comes to the analysis of deep neural networks (Nagarajan & Kolter, 2019b). The core component of a generalization bound is a *complexity measure*, a quantity that relates to some aspect of generalization. A complexity measure may depend on the properties of the trained model, optimizer, and possibly training data, as long as it does not have access to a validation set. The most classic bounds, such as VC-dimension (Vapnik, 1971), are often related to some form of parameter counting which is often too pessimistic for deep neural networks. Norm-based bounds usually rely on the margin and some norms of the model weights (Langford & Caruana, 2001; Bartlett et al., 2017; Neyshabur et al., 2015; 2017b), but these bounds have been ineffective at studying generalization of deep learning (Nagarajan & Kolter, 2019a). Another main class is the PAC-Bayes bounds (McAllester, 1999) which have been much more successful in deep learning due to the flexibility of prior (Neyshabur et al., 2017a; Dziugaite & Roy, 2017; Zhou et al., 2019; Lotfi et al., 2022), although these bounds are still much looser than the actual generalization error. Our approach also belongs to the PAC-Bayes family, but we apply the PAC-Bayes bounds to the distribution of *discrete tokens* (with a language model as the prior) rather than to a distribution over the parameters of a neural network. This allows us to derive significantly tighter bounds compared to applying the PAC-Bayes bounds with less informative priors.

## 3 PRELIMINARIES

**Notations.** Let $\mathcal{X} \in \mathbb{R}^d$ be a set of inputs and $\mathcal{Y} = [K]$ be a label set, and there exists a probability distribution $D$ on $(\mathcal{X} \times \mathcal{Y})$ which is unknown. Let our data $(X_1, Y_1), \ldots, (X_n, Y_n)$ be drawn i.i.d from $D$, and consider a predictor $f : \mathcal{X} \to \mathcal{Y}$ and a fixed set of predictors indexed by the parameter set $\Theta$. We use $f_\theta$ to denote the classifier indexed by $\theta$. We consider the 0–1 loss given by $\ell(y', y) = \mathbb{1}\{y \neq y'\}$. The generalization error (risk) of a predictor is defined as $R(\theta) = \mathbb{E}_{(X,Y) \sim P}[\ell(f_\theta(X), Y)]$ and the empirical risk $r(\theta) = \frac{1}{n} \sum_{i=1}^{n} \ell(f_\theta(X_i), Y_i)$ satisfies $\mathbb{E}_\mathcal{S}[r(\theta)] = R(\theta)$ for a sample $\mathcal{S} = [(X_1, Y_1), \ldots, (X_n, Y_n)]$. An estimator is a function $\hat{\theta} : \bigcup_{n=1}^{\infty} (\mathcal{X} \times \mathcal{Y})^n \to \Theta$.

**Vision-language models.** CLIP consists of two encoders $\text{enc}_\text{img}$ and $\text{enc}_\text{txt}$. Given an image $X \in \mathcal{X}$, the image encoder $\text{enc}_\text{img} : \mathcal{X} \to \mathbb{R}^d$ maps an image $X$ to a $d$-dimension real-valued embedding. Let $\mathcal{T}$ be the space of texts and $T \in \mathcal{T}$ a single piece of text, the image encoder $\text{enc}_\text{txt} : \mathcal{T} \to \mathbb{R}^d$ maps $T$ to a $d$-dimension real-valued embedding. Given a batch of images $\{X_i\}_{i=1}^{B}$ and their corresponding texts $\{T_i\}_{i=1}^{B}$, the training objective maximizes the cosine similarity of the embeddings of the matching image and text pair and minimize the cosine similarity of image and text pairs that do not correspond to each other. The primary task we consider in this work is image classification via pretrained vision-language models. The goal is to find a *class prompt*, $\theta^k \in \mathcal{T}$, for each class that achieves good accuracy. For a $K$-class classification problem with $\theta = (\theta^1, \theta^2, \ldots, \theta^K) \in \Theta = \mathcal{T}^K$, the zero-shot classifier is $f_\theta(X) = \arg\max_{k \in [K]} \langle \text{enc}_\text{txt}(\theta^k), \text{enc}_\text{img}(X) \rangle$.

**Generalization bounds.** Deriving generalization bounds is closely related to assigning hypotheses prior probabilities of being good (Shalev-Shwartz & Ben-David, 2014). One of the simplest approaches uses uniform convergence over the entire discrete hypothesis space (where $|\Theta|$ denotes the number of functions in the class) to derive the well-known generalization bound,

**Theorem 3.1** (Shalev-Shwartz & Ben-David (2014))**.** *For every $\delta > 0$, with probability $1 - \delta$ over the training set of size $n$, for any hypothesis $\theta \in \Theta$, the following holds* $R(\theta) \leq r(\theta) + \sqrt{\frac{\log |\Theta| + \log(\frac{1}{\delta})}{2n}}$.

This result does not consider the implicit bias of the learning algorithm (Neyshabur et al., 2014), the training data $\mathcal{S}$, or the data-generating distribution $D$. In contrast, the PAC-Bayes framework offers a flexible approach for leveraging this information by defining a hierarchy over hypotheses in the hypothesis class $\Theta$ that takes the form of a prior distribution $P$ over $\Theta$. That is, we assign a probability $P(\theta) \geq 0$ for each $\theta \in \Theta$ and refer to $P(\theta)$ as the prior score of $\theta$. The learning process defines a posterior probability over $\Theta$, which we denote by $Q$. In the context of supervised learning, we can think of $Q$ as defining the following prediction rule: given an instance $X$, we randomly pick a hypothesis $\theta$ according to $Q$ and predict $f_\theta(X)$. Remarkably, it was shown that the expected generalization gap can be upper bounded by the KL-divergence between $P$ and $Q$:

**Theorem 3.2** (McAllester (1999)). *For every $\delta > 0$, prior $P$ over $\Theta$, with probability $1 - \delta$ over the training set of size $n$, for any posterior $Q$ over $\Theta$, the following holds*

$$\mathbb{E}_{\theta \sim Q}[R(\theta)] \leq \mathbb{E}_{\theta \sim Q}[r(\theta)] + \sqrt{\frac{D_{KL}(Q \,\|\, P) + \log(\frac{n}{\delta}) + 2}{2n - 1}}. \tag{1}$$

## 4 METHODOLOGY

Designing a prompt is analogous to finding a set of weights in typical machine learning models, where the hypothesis space is the space of texts/tokens. The goal is to find class prompts that maximize training accuracy without finetuning the model's parameters. This process, which is often referred to as *prompt engineering*, can be formulated as discrete optimization over the space of tokens, $\mathcal{V}$.

### 4.1 PROMPT SEARCH

To study the generalization capabilities of discrete prompts, we consider a simple greedy search algorithm that mimics an overeager prompt engineer who exhaustively tries adjusting prompts with every possible word, although the analysis extends to other techniques that produce discrete prompts. To find class prompts of length $L$, we will search for $K \cdot L$ tokens over the space, $\mathcal{V}^{K \cdot L}$. Naively, this search is exponential in the length of the prompt so to circumvent this problem, the prompts are generated successively; that is, we increment the prompts by selecting the token that maximizes a **search criterion**, $\mathcal{J}$, on the training dataset from a set of **candidate tokens**, $\widehat{\mathcal{V}} \subseteq \mathcal{V}$. With a slight abuse of notation, we will use $\widehat{\mathcal{V}}(\theta)$ to denote a candidate set that can be conditioned on the current $\theta$.

The search criterion is the objective being optimized (e.g., the empirical loss), and candidate tokens are permissible tokens that can be used to extend the current class prompts. At every step of the search, we keep the class prompts fixed except for all but one class. The prompt for each class $k$ is a sequence of $l$ tokens $\theta_l^k \in \mathcal{V}$, $\theta_{\leq l}^k = (\theta_1^k, \theta_2^k, \ldots, \theta_l^k)$ where $l < L$, and we use $\theta^{\neg k}$ to denote the class prompts for all classes that are not the $k^{\text{th}}$ class. The next token for $\theta^k$ is obtained via:

$$\theta_{l+1}^k = \underset{v \in \widehat{\mathcal{V}}(\theta)}{\arg \max} \; \mathcal{J}\big(v, \theta_{\leq l}^k, \theta^{\neg k}\big). \tag{2}$$

The pseudocode for this sequential search is outlined in detail in Algorithm 1.

**Empirical risk minimization.** Using $\oplus$ to denote concatenation, we consider a simple form of search, *greedy search*, where we use:

$$\widehat{\mathcal{V}}_{\text{greedy}}(\theta) = \mathcal{V}, \quad \mathcal{J}_{\text{greedy}}\big(v, \theta_{\leq l}^k, \theta^{\neg k}\big) = -r\big((\ldots, \theta^{k-1}, \theta_{\leq l}^k \oplus v, \theta^{k+1}, \ldots)\big), \tag{3}$$

where $r$ is the empirical risk in terms of the 0–1 loss (see Section 3). In other words, we always search over all possible tokens (line 6) to maximize the training accuracy. This greedy search is an *empirical risk minimization* (Vapnik, 1991, ERM) learner since its only objective is to minimize the training error. There are several drawbacks to this simple algorithm, the chief of which is that we need to search over $\mathcal{V}$ exhaustively at each step, which can be expensive since it consists of all the tokens of the vision-language model (e.g., CLIP has about 50000 tokens). Instead, we could search over only a subset of $\mathcal{V}$. To reduce this search space, we use a language model (LM) to induce a distribution over the next tokens conditioned on $\theta^k$ and only evaluate the tokens with high probabilities:

$$p_{\text{next}}(\theta_{l+1}^k \mid \theta_{\leq l}^k) = p_{\text{LM}}\big(\theta_{l+1}^k \mid \theta_{\leq l}^k = (\theta_1^k, \theta_2^k, \ldots, \theta_l^k)\big). \tag{4}$$

Given that CLIP is trained with natural language supervision, autoregressive LMs that are also trained on natural language can likely predict suitable next tokens. We then take the top $N$ candidates and only evaluate the accuracy of these candidates. Conveniently, this can be seen as constraining the complexity of the prompt as the language model provides a structured prior. We observe that this pruning incurs minimal performance loss, suggesting that LMs indeed are good prior in searching for class prompts on image classification tasks. Furthermore, we may use predefined strings to further constrain the hypothesis space by starting with an *initial prompt* such as "This is an image of [...]", instead of an empty string. These initial prompts can provide additional structure to the generated prompts by constraining the output distribution, similar to the role of inductive bias. We refer to this method as Greedy.

**Structural risk minimization via PAC-Bayes.**     This procedure can be further augmented to optimize the PAC-Bayes bound via *structural risk minimization* (Vapnik & Chervonenkis, 1974, SRM) similar to the approach of Dziugaite & Roy (2017), namely, we will take the hypothesis complexity (e.g., KL-divergence) into account as we search for the next token for each prompt. We use the KL-divergence directly in the objective optimization without sacrificing the quality of the solution. Once again, we do this optimization in a sequential manner via Algorithm 1:

$$\widehat{\mathcal{V}}_{\text{LM}}(\theta) = \left\{ v \in \mathcal{V} \,\Big|\, \max_{v'} p_{\text{next}}(v' \mid \theta_{\leq l}^k) - p_{\text{next}}(v \mid \theta_{\leq l}^k) \leq \Delta \right\}, \tag{5}$$

$$\mathcal{J}_{\text{LM}}(v, \theta_{\leq l}^k, \theta^{\neg k}) = -r\left((\ldots, \theta^{k-1}, \theta_{\leq l}^k \oplus v, \theta^{k+1}, \ldots)\right) + \beta \log p_{\text{next}}(v \mid \theta_{\leq l}^k), \tag{6}$$

where $\Delta$ controls the size of the search space (adjusted according to computational constraints) and $\beta$ is a hyperparameter that controls the strength of the regularization. This set of permissible tokens could also be pruned and fixed beforehand by discarding tokens with low marginal probability. We refer to this version of search as regularized greedy.

## 4.2 Generalization Guarantees for Prompts

Since the space of all prompts is discrete and the total number of possible prompts is $|\Theta| = |\mathcal{V}|^{LK}$, for a single hypothesis $\hat{\theta}$, we have the following uniform convergence bound for prompts that depends on prompt length, the number of classes, and the number of tokens in the vocabulary by assigning uniform probability to each hypothesis (from Theorem 3.1):

$$R(\hat{\theta}) \leq r(\hat{\theta}) + \sqrt{\frac{L\,K\,\log|\mathcal{V}| + \log(\frac{1}{\delta})}{2n}}. \tag{7}$$

However, not all prompts are equally likely to be good. To obtain a tighter generalization guarantee on the learned $\hat{\theta}$, we will leverage a classical PAC-Bayes bound to derive an upper bound on the generalization error of the learned prompts.

In conventional application of PAC-Bayes to deep learning, $P$ and $Q$ are often chosen to be isotropic Gaussian on the parameters (Langford & Caruana, 2001) so the KL-divergence between the prior and posterior can be easily computed. We instead use a language model as the prior over $K$ independent prompts, $P(\theta) = \prod_{i=1}^K \prod_{j=1}^L p_{\text{LM}}(\theta_j^i \mid \theta_{\leq j}^i)$. Further, we treat the prompts $\hat{\theta}$ found through search or through prompt engineering as a point mass posterior, $Q(\theta) = \mathbb{1}\{\theta = \hat{\theta}\}$. In this case, the KL-divergence is conveniently equal to the negative log-likelihood of $\hat{\theta}$ under the LM because the posterior is zero everywhere except for at $\hat{\theta}$:

$$D_{\text{KL}}(Q \parallel P) = \sum_{\theta \in \Theta} Q(\theta) \log \frac{Q(\theta)}{P(\theta)} = \log \frac{1}{P(\hat{\theta})} = -\sum_{i=1}^K \sum_{j=1}^L \log p_{\text{LM}}\left(\hat{\theta}_j^i \mid \hat{\theta}_{\leq j}^i\right). \tag{8}$$

This bound has an intuitive interpretation, which is that the generalizing prompts are the ones that achieve good training performance and are likely under the language model. Having a point-mass posterior over discrete space also means that we can *derandomize* the PAC-Bayes bound for free (Viallard et al., 2021). Combining these observations, we have the following deterministic upper bound on the generalization error (from Theorem 3.2):

$$R(\hat{\theta}) \leq r(\hat{\theta}) + \sqrt{\frac{-\sum_{i=1}^K \sum_{j=1}^L \log p_{\text{LM}}\left(\hat{\theta}_j^i \mid \hat{\theta}_{\leq j}^i\right) + \log(\frac{n}{\delta}) + 2}{2n - 1}}. \tag{9}$$

Table 2: Performance and generalization bounds for prompts produced by `Greedy` and for hand-crafted prompts on different datasets with different CLIP architectures. UC represents the uniform convergence bound. Handcrafted prompts are taken from CLIP and Wise-FT (Wortsman et al., 2022).

| Dataset | Model | Method | Train Err | Test Err | UC | PAC-Bayes |
|---------|-------|--------|-----------|----------|-----|-----------|
| CIFAR-10 | B-16 | `Greedy` | 0.050 | 0.060 | 0.154 | 0.086 |
| | L-14 | `Greedy` | 0.023 | 0.028 | 0.128 | 0.063 |
| | L-14 | `handcrafted` | 0.040 | 0.040 | 0.145 | 0.078 |
| CIFAR-100 | B-16 | `Greedy` | 0.208 | 0.255 | 0.537 | 0.317 |
| | L-14 | `Greedy` | 0.142 | 0.180 | 0.471 | 0.266 |
| | L-14 | `handcrafted` | 0.221 | 0.221 | 0.549 | 0.339 |
| fMoW | B-16 | `Greedy` | 0.598 | 0.621 | 0.807 | 0.667 |
| | L-14 | `Greedy` | 0.514 | 0.547 | 0.723 | 0.596 |
| | L-14 | `handcrafted` | 0.725 | 0.402 | 0.934 | 0.804 |
| OfficeHome | B-16 | `Greedy` | 0.104 | 0.150 | 0.635 | 0.281 |
| | L-14 | `Greedy` | 0.070 | 0.115 | 0.601 | 0.260 |
| | L-14 | `handcrafted` | 0.926 | 0.928 | 1.457 | 1.119 |
| ImageNet | L-14 | `handcrafted` | 0.243 | 0.256 | 0.448 | 0.319 |

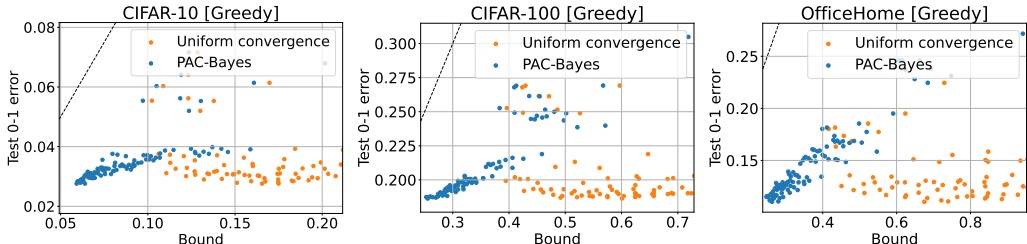

Figure 1: Test error vs generalization bound on CIFAR-10, CIFAR-100, and OfficeHome. We compare the uniform convergence bound and PAC-Bayes bound, when evaluated on prompts produced by `Greedy`. The dashed line represents $y = x$.

We note that these techniques are not novel from a theoretical perspective and there are more sophisticated PAC-Bayes variants that may yield tighter results. Nonetheless, in the next section, we will observe that this simple bound is *surprisingly* tight even for complex datasets such as ImageNet.

**Data leakage and contamination.** One strong assumption of these bounds, which we make explicitly and which could indeed be violated in practice, is that the image encoder is trained without access to the training set used for prompt engineering. If it is trained on this data, even from the *training* set, then the functional complexity of the hypothesis class depends not just on the prompt, but also implicitly on the complexity of the image encoder. We emphasize that this fact does not change the nature of the bounds above, but it *does* change whether or not any given bound in the experiments can be formally considered a valid bound, or could be violated.

In practice, this is difficult to verify for the e.g. CLIP encoder, since the data it was trained on is not publicly disclosed. Nonetheless, the CLIP paper includes a sensitivity analysis that shows a relatively small effect of including any of the evaluation datasets they consider (Radford et al., 2021). Thus, while we fully acknowledge that data contamination may apply to the experiments below, we believe this to be similar to many current evaluations of foundation models, where it is difficult to assess the extent to which any performance is truly zero-shot.

## 5 EXPERIMENTS

In this section, we evaluate the generalization of discrete prompts generated by `Greedy` on CIFAR-10, CIFAR-100, ImageNet as well as domain generalization datasets fMoW (Christie et al., 2018) and OfficeHome (Venkateswara et al., 2017), which is much less studied in the context of numerical

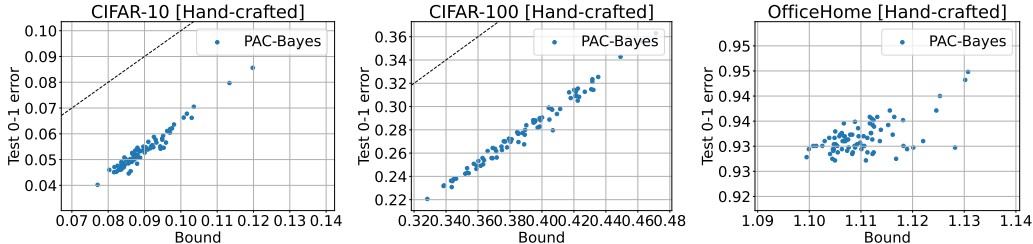

Figure 2: Test error vs PAC-Bayes generalization bound on CIFAR-10, CIFAR-100, and OfficeHome on handcrafted prompts. The dashed line represents $y = x$.

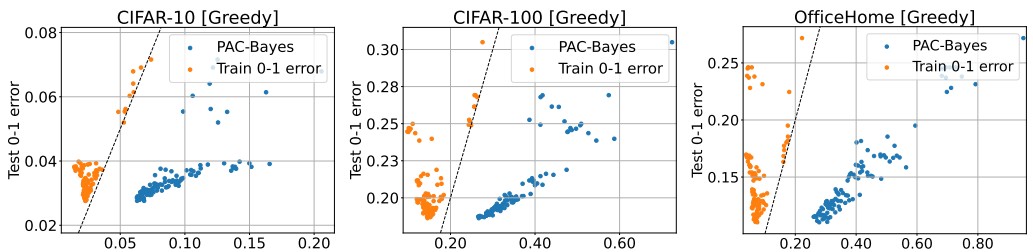

Figure 3: Train error (orange) and generalization bound (blue) vs test error (y-axis) on CIFAR-10, CIFAR-100, and OfficeHome of prompts produced by Greedy. The dashed line represents $y = x$. Notice that towards the region of low training loss (left), many prompts actually have *higher* test loss (negative correlation). On the other hand, the low bounds correlate with low test errors well.

generalization bounds. We also evaluate existing well-performing handcrafted prompts taken from CLIP and Wise-FT (Wortsman et al., 2022). Given these prompts, we compute generalization bounds via PAC-Bayes (PAC-Bayes) and via uniform convergence (UC). The PAC-Bayes bounds are computed using LLaMA-7B (Touvron et al., 2023) as the prior. Within Greedy, we search using the CLIP vocabulary of 49 408 tokens and measure the generalization bounds for 100 realizations of Greedy with each corresponding to a fixed prompt length $l \in \{1, \ldots, 10\}$ and split portion of the dataset $s \in \{0.1, \ldots, 1.0\}$. More details on the experimental procedure are in Appendix C.

**Baselines** We compare our generalization bounds against existing generalization bounds on CIFAR-10, CIFAR-100, and ImageNet. In particular, we compare against the works of Lotfi et al. (2022) and Zhou et al. (2019), which represent the latest progress in PAC-Bayes bounds for deep learning.

As shown in Table 1, discrete prompts achieve much tighter bounds than the state-of-the-art across all 3 datasets. We remark that our approach is also data-independent, while still achieving a tighter bound than the data-dependent approach in the work of Lotfi et al. (2022). An added benefit of this result is that we make little modification to the existing learning paradigm – indeed prior bounds often need to make strict assumptions about the neural network such as Gaussian posterior or the weights lying in a low dimensional manifold (Lotfi et al., 2022) which may hurt the performance.

We observe that even simple UC bounds over discrete prompts generated by Greedy lead to tight, non-vacuous

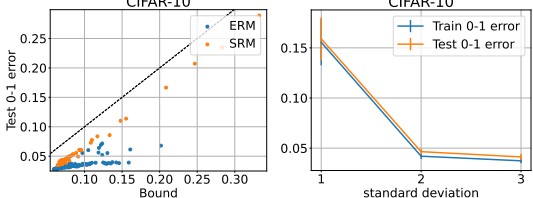

Figure 4: Test error vs the PAC-Bayes bound on CIFAR-10 when using SRM (i.e., directly penalizing the PAC-Bayes bound) (left). We also report the train and test performance when the CLIP vocabulary is pruned (i.e., removing tokens that have logit values that are $k$ standard deviations away from the max token) using the language model (right). This yields prompts with tighter bounds at the cost of slightly higher error.

bounds across a variety of datasets, and PAC-Bayes bounds with an LLM prior further improve these bounds (Figure 1). These also apply to handcrafted prompts (Figure 2) from the existing literature (Radford et al., 2021; Wortsman et al., 2022) (other datasets' result in Appendix B).

**Structural risk minimization with the PAC-Bayes bound**    PAC-Bayes is related to SRM (Vapnik & Chervonenkis, 1974), where one tries to optimize both the goodness of fit and complexity of the model. When we compare test error against train error or the generalization bound (Figure 3), we observe that the generalization bound can serve as a useful criterion for model selection. We consider using SRM, where our complexity term is exactly the KL divergence term in Equation 8. `Regularized Greedy` now jointly maximizes train accuracy and minimizes this KL divergence term when adding new tokens to each class prompt. We observe that this naturally leads to tighter bounds for prompts yielded by `Greedy` on CIFAR-10 (Figure 4) while maintaining comparable accuracy. Interestingly, using LLaMA-7B as the prior does not significantly improve the linguistic coherence of prompts obtained through regularized search, which leaves room for more sophisticated search techniques to address this in future work.

**Pruning the hypothesis space**    In addition to regularizing the search objective with the KL term directly, another method to improve our generalization bounds is to prune the vocabulary using a large language model. We experiment with conditioning the language model on the class names and then selecting tokens from the language model's vocabulary with the highest probability under the language model. In Figure 4, we report the performance and generalization of `Greedy` when the tokens considered in search are restricted to within $k$ standard-deviations (see Appendix B.1 for details) away from the maximum logit token. While the vocabulary size of LLaMA-7b is $32\,000$ tokens, the number of tokens within 3, 2, 1 standard deviations from the maximum token are 6894, 1361, 185 respectively. We observe this implicitly prunes the hypotheses to contain those with smaller generalization error at a small cost to the train and test error. Restricting the vocabulary also encodes prior knowledge about the data or domain. For example, further results using a vocabulary of English words in Appendix B.1 (instead of CLIP's vocabulary of tokens) show that we can learn slightly more interpretable prompts.

**Effects of prompt length** Another key quantity of prompt engineering is the prompt length which directly controls the size of the hypothesis space. We analyze how the length of class prompts impacts the performance of `Greedy` (Figure 5). We note that at a certain length, the train accuracy plateaus, which means that a relatively small prompt length suffices for good classification performance.

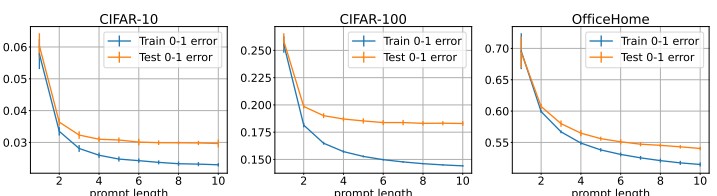

Figure 5: The train and test accuracy with different prompt lengths for greedy search. Although the generalization gap increases with prompt length, there is little overfitting even at the longest lengths.

**Fitting random labels**    Motivated by our new observations about prompt engineering, we hypothesize that the learned prompts are less prone to overfitting the noise in the data. Zhang et al. (2021) showed that conventional deep neural networks can fit both *random labels*, arguing that these models have much higher capacity than what traditional statistical learning theory can deal with. To demonstrate that prompt engineering is robust to label noise, we experiment with running `Greedy`

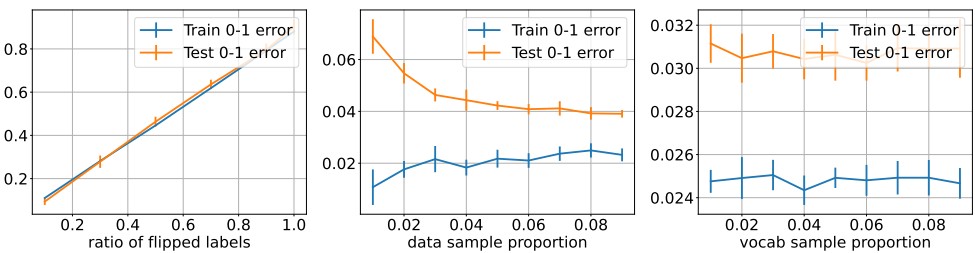

Figure 6: We show the generalization of discrete prompts produced by `Greedy` on randomly labeled data from CIFAR-10 (left). We also report the performance when search is done with 1% - 9% of the labeled data (middle), and when search is done with 1% - 9% of the CLIP vocabulary (right). We fix the prompt length to be 5.

Table 3: Performance and generalization bounds for prompts produced by `Greedy` and for a linear probe (on top of CLIP features) on different datasets with 20 samples per class. UC represents the uniform convergence bound. We omit UC for linear probing because this is a multi-class problem.

| Dataset | Model | Method | Train Err | Test Err | UC-20 | PAC-Bayes-20 |
|---------|-------|--------|-----------|----------|-------|--------------|
| CIFAR-10 | L-14 | `Greedy` | 0.020 | 0.138 | 1.675 | 0.634 |
| | L-14 | `Linear Probe` | 0.000 | 0.038 | - | 2.591 |
| CIFAR-100 | L-14 | `Greedy` | 0.156 | 0.367 | 1.801 | 0.637 |
| | L-14 | `Linear Probe` | 0.000 | 0.198 | - | 3.715 |

on training data with a certain proportion of randomly flipped labels. We observe that both training and test accuracy drop monotonically in tandem as we flip these training labels (Figure 6), which suggests that the prompts cannot overfit the random labels. For a baseline comparison, we also compare the performance of a linear probe on random labels. We observe that this achieves roughly random performance (13.60% accuracy) with 100% flipped labels. This supports that Greedy is not too simple of a search approach to fit the random labels as other more complex methods also cannot.

**Learning with small data**  When the number of data points is small (e.g., $n = 20$), the use of PAC-Bayes is especially attractive since we can use all the data points to estimate the posterior and bound its risk. Furthermore, prompt engineering is frequently used with limited labeled data; thus, further progress in understanding its generalization properties must provide bounds in this regime. In Figure 6, we report the train and test accuracy of `Greedy` as we vary the amount of training data (between 1%–10% of the full data) we use in computing the search objective. We observe less than 2% increase in error with 2% of the training set of CIFAR-10. This highlights that `Greedy` can be remarkably data efficient. We then compute both the uniform convergence and PAC-Bayes bounds with 20 samples per class (Table 3). The results underscore the importance of an informative prior in the form of the LLM. The bounds obtained with the LLM prior are, albeit loose but still non-vacuous. To the best of our knowledge, this is not possible with prior approaches unless it is data-dependent. One could ask since we assume the representation from CLIP is not learned from the training data, can we simply use an SVM-like bound on the learned features (McNamara & Balcan, 2017)? As a case in point, we present a standard linear probe (on top of CLIP's features), which achieves slightly better accuracy but a vacuous generalization bound. The implementation details are described in Appendix C. The discrete nature of prompts and the fact that the corresponding hypothesis space of CLIP is so small is crucial to the success of our approach. We believe that exploring avenues to obtain tighter PAC-Bayes bounds in the small data regime is an opportunity for future work and the use of data-dependent priors may be fruitful in this regard.

## 6 CONCLUSION AND LIMITATIONS

In this paper, we study the generalization properties of engineered prompts on image recognition tasks. We observe the surprising fact: prompt engineering does not seem to overfit, and also performs well on the test distribution. We provide a principled approach to analyze this generalization behavior by framing discrete prompts as a relatively small hypothesis class, onto which we can naturally apply classical PAC-Bayes bounds using an LLM prior. This results in the *tightest* bounds yet observed across multiple complex datasets, including CIFAR-10, CIFAR-100, and ImageNet. As a whole, this supports the use of prompt-engineering or simple greedy searches over potential class prompts as a high-performing and well-generalizing classifier.

Despite the ability to produce highly non-vacuous bounds, the bounds rely on the fact that pretrained vision-language models readily contain some hypothesis class that will perform well on the training set (for whatever the desired task is). This, in turn, naturally relies on the generalization performance of the underlying model itself, which our analysis evidently does not, and cannot, address (as they are only aware of the language model, which does not observe the data).

Nonetheless, what our bounds *do* address is the fact that when *given* these performant models, manual prompt engineering (even when "overfitting" to a training set) often exhibits *surprisingly* strong generalization behavior. Given the prevalence of prompt engineering in modern ML, we believe that this work provides an important perspective on this widespread practice.

## ACKNOWLEDGEMENTS

We thank Nina Balcan for valuable discussions during this project. Victor Akinwande, and Dylan Sam were supported by funding from Bosch Center for AI. Dylan Sam was also supported by a National Science Foundation Graduate Research Fellowship under Grant No. DGE2140739 and the ARCS Foundation. Yiding Jiang is supported by the Google PhD Fellowship.

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

## A PESUDOCODE

---

**Algorithm 1** Sequential Prompt Search

---

1: $\theta \leftarrow$ [initial_prompt]$\times K$
2: **for** $l = 0$ to $L - 1$ **do**
3:     class_order $\leftarrow$ randomly sampled order of class indices
4:     **for** $k$ in class_order **do**
5:         criteria $\leftarrow -\infty$
6:         **for** $v$ in $\widehat{\mathcal{V}}(\theta)$ **do**                ▷ This step is vectorized in practice.
7:            score $\leftarrow \mathcal{J}(v, \theta_{\leq l}^k, \theta^{\neg k})$                   ▷ Evaluate the score of $v$.
8:            **if** score > criteria **then**         ▷ Keep the prompt with best performance.
9:                criteria $\leftarrow$ score                ▷ Update the current best score.
10:                $\theta_{l+1}^k \leftarrow v$                     ▷ Update $\theta^k$ with the better token.
11: **Return** $\theta$

---

## B ADDITIONAL RESULTS

**Other datasets** We report the results of discrete prompts both generated by `Greedy` and hand-crafted on FMoW in Figure 7.

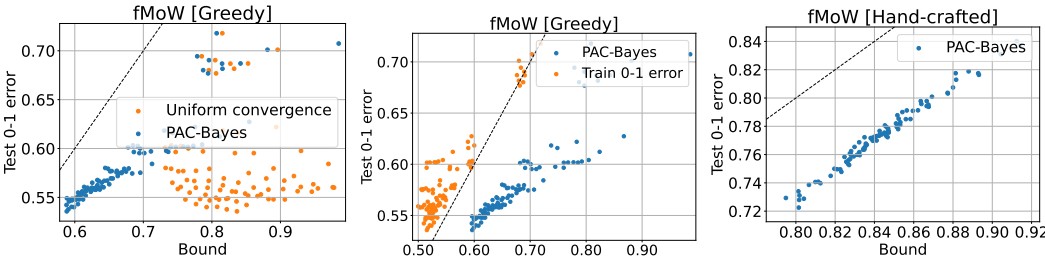

Figure 7: Test error vs generalization bounds on fMoW. We report the uniform convergence bound and PAC-Bayes bound when evaluated on prompts produced by `Greedy` (left). We plot its train vs. test error (middle). We also report the performance of handcrafted prompts and their corresponding PAC-Bayes bound (right). The dashed line is $y = x$.

**Creating the pruned vocabulary** To create the pruned vocabulary, we take the class name of each class (e.g., "dog") and feed each one into the LLM and compute the logits over the next token. We then compute the standard deviation, $\sigma$, over each class's logits and take the tokens that are $k\sigma$ from the maximum logit. Then, we use the union of the top tokens of all classes as the pruned vocabulary.

**Vocabulary subsampling** In addition to the result on CIFAR-10 in Figure 6, we report the performance of discrete prompts generated by greedy on CIFAR-100, when a random subset of the CLIP vocabulary is used in Figure 8. We observe less than 2% increase in error with 1% of the vocabulary. This provides further evidence of the robustness of `Greedy` to the vocabulary size. Random sampling, while easy to implement, prunes hypotheses that may have desirable properties. As such, we report the performance of greedy on CIFAR-100 when the vocabulary is pruned using the language model, and observe that `Greedy` can recover prompts with better generalization (See Figure 9).

**Fitting random labels** In addition to the result on CIFAR-10 in Figure 6, we report results on fitting to randomly labeled data for CIFAR-100, FMoW, and OfficeHome in in Figure 10, and observe consistently that `Greedy` does not fit random labels. This provides evidence that contrasts the current literature on the ability of neural networks to easily fit random labels.

**Fitting with small data** In Figure 11 we report results on fitting to small sample sizes on both CIFAR-10 and CIFAR-100. We consider random subsets between 1% – 10% of the data and between

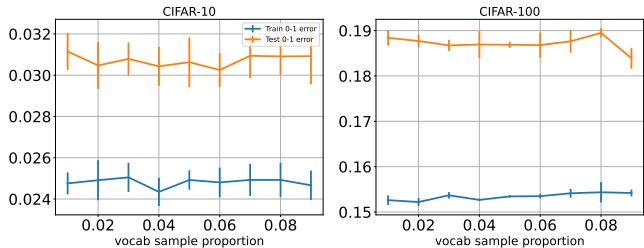

Figure 8: We show the generalization of `Greedy` when search is done with 1% - 9% of the tokens sampled randomly from the CLIP vocabulary on CIFAR-10 (left), and CIFAR-100 (right). We fix the prompt length to be 5.

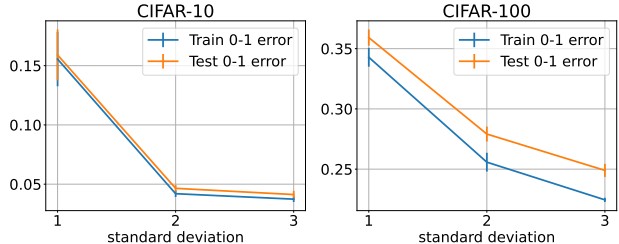

Figure 9: We show the generalization of `Greedy` when search is done with subsets of the tokens sampled from the language model as described in the text on CIFAR-10(left), and CIFAR-100(right). We fix the prompt length to be 5.

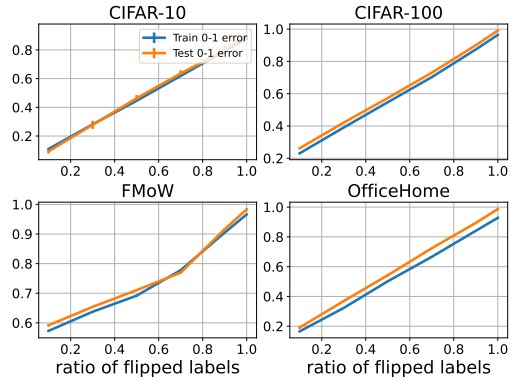

Figure 10: We show the generalization of `Greedy` with randomly labeled data on CIFAR-10, CIFAR-100, FmoW, and OfficeHome. We fix the prompt length to be 5.

0.1% − 1%. We observe that the discrete prompts that `Greedy` can learn, even with small sample sizes, observe good generalization that degrades as the sizes of the training set decrease.

**Learning with a different vocabulary**    The `Greedy` algorithm is agnostic to the set of tokens used in the search procedure. In practice, one may use a vocabulary that encodes prior knowledge about the data or domain. Additionally, certain properties like interpretability may be desired. We report results on searching with the language model's vocabulary (See Figure 12). We do not observe significant degradation in performance. We also report results on penalizing the search criteria using the bound (i.e. SRM) with different $\beta$ values (See Figure 13). We observe that `Greedy` is able to recover prompts with better generalization as the penalty increases at a small cost to accuracy. We run `Greedy` on a vocabulary of English words obtained from the *english-words*[2] package. We show the prompts learned on CIFAR-10 in Figure 15.

---

[2] https://github.com/mwiens91/english-words-py

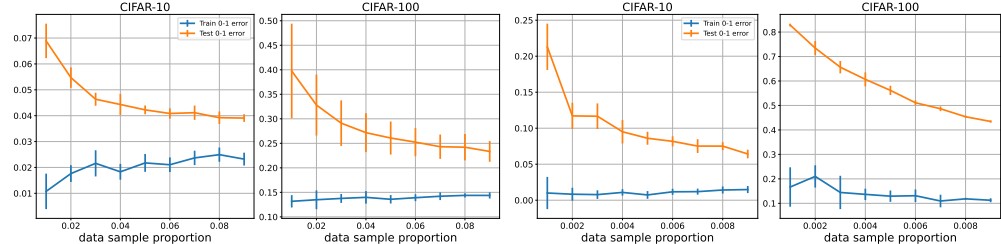

Figure 11: We show the generalization of `Greedy` when search is done with 1% - 9% of the data sampled randomly on CIFAR-10 (left), and CIFAR-100 (second-left). We also show the generalization of `Greedy` when search is done with 0.1% - 0.9% of the data sampled randomly on CIFAR-10 (second-right), and CIFAR-100 (right). We fix the prompt length to be 5.

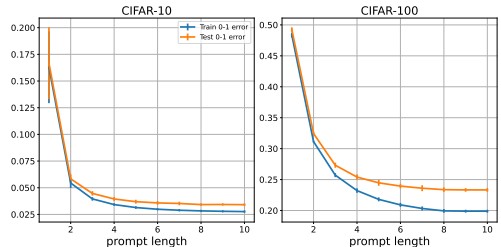

Figure 12: We show the generalization of `Greedy` with the Llama-7b vocabulary on CIFAR-10 (left) and CIFAR-100 (right).

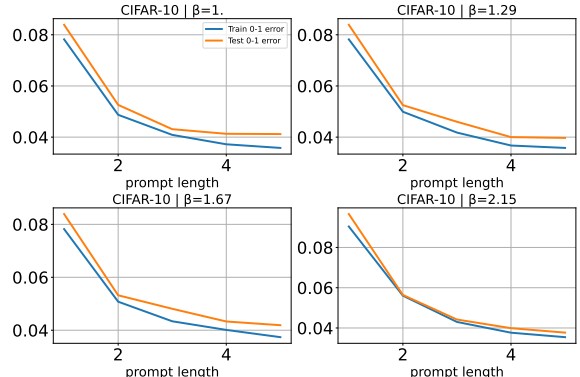

Figure 13: We show the generalization of `Greedy` with the LLaMA-7b vocabulary on CIFAR-10 with different values of penalty $\beta$ with the SRM objective.

## B.1 PROMPTING THE LANGUAGE MODEL BEFORE SEARCH

**Prompt prefixes**    We prefix `Greedy` with:

1. `a photo of a`
2. `a blurry photo of a`
3. `a black and white photo of a`
4. `a low contrast photo of a`
5. `a high contrast photo of a`
6. `a bad photo of a`
7. `a good photo of a`
8. `a photo of a small`

9. `a photo of a big`

10. `a photo of the`

We show the performance on CIFAR-10 in Figure 14. We observe that `Greedy` consistently converges to a prompt with roughly the same test-error regardless of the prefix.

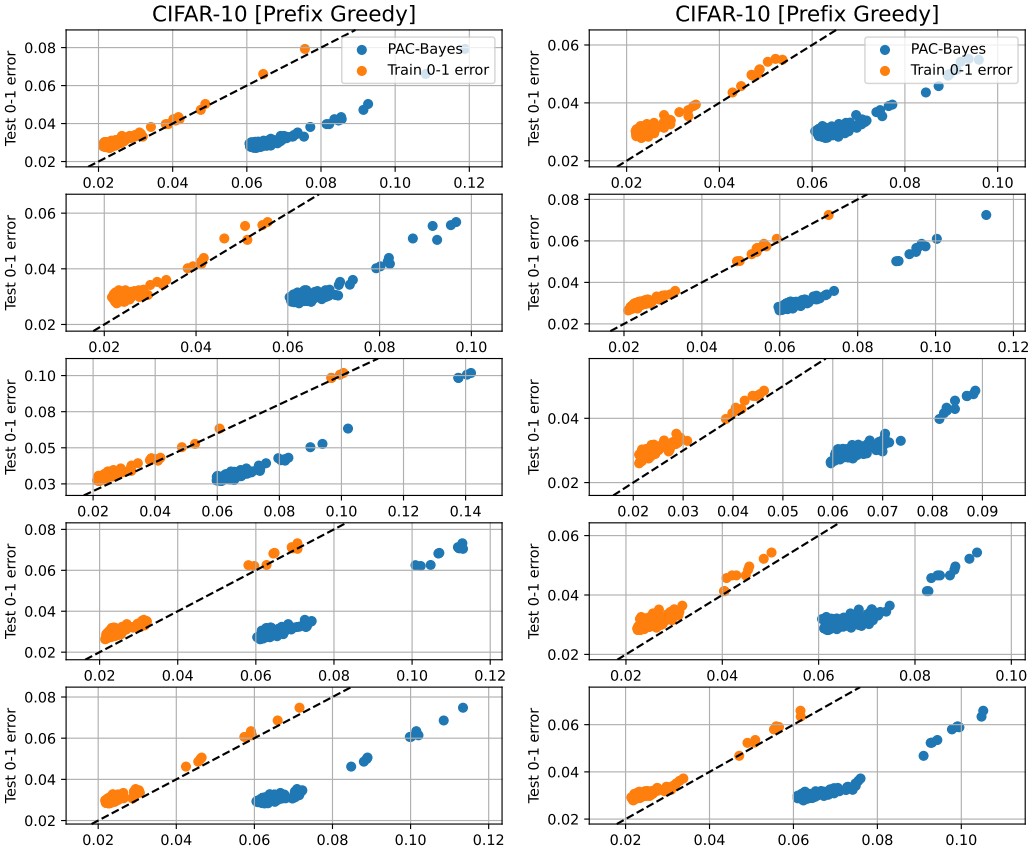

Figure 14: When Greedy is prefixed with a set of tokens, it consistently converges to prompts with with roughly the same test-error regardless of the prefix.

```
[airplane, automobile, bird, cat, deer, dog, frog, horse, ship, truck]

aviarist nonsonant confirmment establishmentism hemiteratics
nonmotoring known vaticinal allot nth
ornithophile slimsy renishly redivive muchness
wheencat compearant stintedness osiery thisness
stagnature unchawed lophobranch primariness primariness
dogrib babooism pneumococcic kaoliang bogusness
froggery phthalid auhuhu rippit hideousness
horsemonger fooyoung inordinary spreadingness forthbring
seafaring rumness tralatition babeship knocking
truckling phthartolatrae semantology waywarden decess
```

Figure 15: We show the learned prompts using a full-word vocabulary of English words on CIFAR-10. This achieves 3.3% test error with the L-14 base model.

## C EXPERIMENTAL DETAILS

**Hyperparameters**   We report the hyperparameters used in CLIP, LLaMA-7b, and the `Greedy` algorithm in Table 4.

Table 4: Hyperparameters used in CLIP, LLaMA-7b and `Greedy`.

| Hyperparameter | Value |
|---|---|
| Batch size | 100 |
| CLIP Vocabulary size | 49,408 |
| LLaMA-7B Vocabulary size | 32,000 |
| Temperature | 1.0 |
| Bound $\delta$ | 0.01 |
| SRM $\beta$ | 1.0 |

**Linear Probe Baseline**   A linear probe baseline was trained with a batch size of 64 and a learning rate of 0.01 for 10 epochs. We compute the generalization bound using McAllester's bound with a prior distribution over linear model weights of $\mathcal{N}(w^{(0)}, \sigma^2 I)$ and a posterior of $\mathcal{N}(w, \sigma^2 I)$, where $I$ is an identity matrix of dimension (768 $\times$ the number of classes), $w$ is our learned weights, and $w^{(0)}$ is our random initialization. We then optimize over a grid of 20000 values for $\sigma \in [0.1, ..., 1]$. This mirrors the procedure from the work of Jiang et al. (2019). We also note that computing a standard UC bound is challenging as we cannot specify a meaningful prior over an infinite space. Other approaches are challenging due to difficulties in computing multiclass analogues of the VC dimension such as the Natarajan dimension (Daniely et al., 2015).

