# OpenReview forum: "Understanding prompt engineering may not require rethinking generalization"
_ICLR.cc/2024/Conference — ICLR 2024 poster_

### Official Review · Reviewer_miVF · 2023-10-31

**Soundness:** 3 good
**Presentation:** 3 good
**Contribution:** 3 good
**Rating:** 6
**Confidence:** 4

**Summary:**

This paper provides a PAC-Bayes analysis of prompt engineering and a simple prompt search routine. The paper starts by proposing a simple greedy prompt search strategy that incrementally adds tokens that maximise the reward sequentially with a regularisation in terms of the probability of the next token as predicted by a language model. The authors then remark provide a PAC-Bayes bound for prompts by formulating the prompt search space as a hypothesis, the LM predicted token probabilities as priors and the posterior as a point mass. The authors show that the resulting bounds are surprisingly tight, and explain the observation that prompt engineering does not seem to overfit massively even though it is optimized on the training set only.

**Strengths:**

- This paper provides much-needed theoretical support for the increasingly popular paradigm of prompt engineering, which, as the authors described, is largely empirical up to this point. The use of PAC-Bayes theory in this context is intuitive and insightful and, to my knowledge, novel, and I'd even be a bit surprised that no one has tried this so far.

- The derivation of bound, which is the paper's main contribution, is easy to follow and seems sound. The resulting bound is a massive improvement over the literature (although I am not extremely familiar with the literature in this area -- I defer a more thorough assessment to another, more experienced reviewer), and the experimental validation also provides empirical evidence supporting the authors' argument.

**Weaknesses:**

I'd like to state that I reviewed and participated in the discussion of a previous version of this manuscript at an earlier conference. The critical weakness is that, as the authors acknowledged in the latest manuscript, the validity of almost all derivations hinges on the assumption that there is no data contamination and that the encoder was not trained on the data that it is asked to infer (i.e., the setting is truly zero-shot). As the authors acknowledged, this is something that cannot be fully proved and disproved, partially due to the fact that what was included in the training data is not fully transparent and publicly known.

However, as important as this weakness is, I think the dilemma is at least not only attributable to this work, and given the relative dearth of theoretical works on this area, I still think there is value in this paper to the community. I'd encourage the authors to emphasize more clearly and explicitly the limitations of their work and analysis throughout the paper. A possible remedy is to pre-train and fine-tune a comparable open-source model from scratch where one can fully control what data the model is trained on to investigate the derived bounds; I acknowledge this may require an enormous amount of compute and some qualitative discussions would suffice.

**Questions:**

Please address the concerns above.

--- **Post-rebuttal** ---

I thank the authors for responding to my review, and I remain positive about this work and its value to the community. I will stick to my original rating recommending acceptance (the reason I cannot give a higher rating is that this work, as mentioned in the original review, does have strong assumptions as also acknowledged by the authors).

---

> ### Author Response · Authors · 2023-11-17
> **Response to Reviewer miVF**
>
> Thank you for your continued support for this work!
>
> > data contamination
>
> Since our last discussion, there has been a new work that shows removing imagenet-like data from LAION does not significantly affect the performance of CLIP [1], so we believe that this offers a partial answer to the question. Among other things, the paper showed that even if there is (almost) no data contamination, CLIP would still work well, and we can expect the bound to hold.
> However, we agree that studying data contamination in pretraining datasets will continue to be an important problem moving forward.
>
> **Reference**
>
> [1] Does CLIP's Generalization Performance Mainly Stem from High Train-Test Similarity? Mayilvahanan et al.

---

### Official Review · Reviewer_q9H1 · 2023-10-31

**Soundness:** 4 excellent
**Presentation:** 3 good
**Contribution:** 3 good
**Rating:** 8
**Confidence:** 4

**Summary:**

This paper uses PAC-Bayesian theory to obtain generalization bounds for prompted classifiers, particularly prompted vision-language models like CLIP. By treating the space of (natural language) prompts as a parameter space, and using a generative pretrained language model as a prior over that space, with a posterior as the delta distribution on the prompt selected by some search procedure, very tight generalization bounds can be obtained for the prompted classifier. These bounds are tighter than other PAC-Bayes bounds in the literature, which were obtained for different model families.

**Strengths:**

- Combining PAC-Bayes bounds with the idea of treating prompts as tunable parameters is an interesting and original idea that seems promising for obtaining tight bounds, since the set of linguistically coherent task-specific prompts is a limited "parameter space", we know from existing empirical results that this set contains examples with low training error, and pretrained language models provide us with a natural choice of prior over this space.

- The bounds obtained by the proposed approach are tighter than the bounds obtained by uniform convergence over the space of possible prompts and tighter than other PAC-Bayes bounds in the deep learning literature, including data-dependent ones (though those papers all consider different hypothesis classes).

- Interesting results on non-vacuous generalization bounds for prompts in the very-low-data setting (20 samples per class). Since prompts are a hypothesis space with a strong prior, this model family seems like a very promising candidate for obtaining strong generalization bounds even in the few-shot-learning regime, which has been largely out of reach. These results push us towards such bounds.

**Weaknesses:**

- The proposed bounds are only evaluated on image classification tasks, but prompting and prompt engineering are equally if not more common in NLP for text classification and other tasks.

- The abstract claims that "the bound is remarkably suitable for model selection: the models with the best bound typically have the best test performance." Based on Figure 3, it seems like the bound is no more useful than just looking at the training error. But it's hard to tell because the points in the plot are not paired. Are there examples where using the PAC-Bayes bound leads to a different and better model selection choice than just using the training error?

- _extends to the methods proposed in Wen et al. (2023)._ That paper has a much more sophisticated search method than the greedy search here. How do we know the resulting prompts are competitive with other search methods? It's less interesting if we're getting generalization bounds for a suboptimal classifier.

- >both training and test accuracy drop monotonically in tandem as we flip these training labels (Figure 6), which suggests that the prompts cannot overfit the random labels.

    - Either that, or the greedy search procedure just can't find a good enough prompt. As far as I can tell, we can't decide based on the evidence in the paper.

- As far as I can tell, the analysis doesn't actually suggest a useful practical algorithm. Based on Figures 3 and 4, if I want to find the best prompt, it seems like my best bet is still to just pick the one that minimizes the training error (using whatever search method I come up with) and not bother with SRM.

**Questions:**

- Does prompting the language model differently induce a better prior? For example, if I prompt GPT-4 with "I'm trying to prompt a vision-language model like CLIP to classify between cats and dogs. Can you suggest a good prompt?" it replies: "This image contains a {cat/dog}." So intuitively it seems like using the likelihood of possible CLIP prompts under this LM prompt would give a much better prior than off-the-shelf LLaMA, and it might also help with fluency. In general, it would've been interesting to see more ablation on the prompt used for the prior.

---

> ### Author Response · Authors · 2023-11-17
> **Response to Reviewer q9H1**
>
> Thank you for your thorough and insightful feedback! We are glad that you find our idea original and results interesting. We will address your concerns as follows:
>
> > The proposed bounds are only evaluated on image classification tasks
>
> This is a good point. This choice was in line with the existing literature on generalization bounds in deep learning. Extending this methodology to NLP is indeed an interesting future direction, though it would require careful consideration of specific protocols since there is not a well-established one.
>
> > Based on Figure 3, it seems like the bound is no more useful than just looking at the training error.
>
> This might be a misunderstanding. We do observe that, especially in regions of low training error, lower PAC-Bayes bounds (blue dots) often align with lower test errors. Conversely, low training errors (orange dots) can correspond to higher test errors. This indicates that our PAC-Bayes bound is a more reliable indicator of generalization than training error alone. We are currently revising the figure's caption to make this point clearer.
>
> >  That paper has a much more sophisticated search method than the greedy search here..
>
> We want to clarify that the goal of our paper is not to propose the best search algorithm (in fact, the one we employ is quite naive although it works really well).
> We acknowledged that the proposed algorithm is simple and is designed in a way that should intuitively overfit. The key observation is that even prompts found in this manner *do not* overfit. Furthermore, the bound is not specific to our search algorithm, it can also be applied to hand-crafted prompts (which we show in the paper) and discrete prompts found via other methods as well, regardless of the search procedure.
>
> > Either that, or the greedy search procedure just can't find a good enough prompt…
>
> Note that the greedy search can find good prompts for non-randomized labels, so this demonstrates that the greedy search is at least a reasonable prompt optimization procedure. For practical reasons, we cannot exhaustively search every possible prompt. However, since the greedy search procedure is designed to only maximize the training objective (in this case fitting a random label), the difficulty of finding a good hypothesis that fits random labels serves as a good indication that the hypothesis space is bad for fitting random labels (or the hypotheses that can achieve low error on random labels occupy a very small volume so they are hard to find).
>
> > As far as I can tell, the analysis doesn't actually suggest a useful practical algorithm. Based on Figures 3 and 4, if I want to find the best prompt, it seems like my best bet is still to just pick the one that minimizes the training error
>
> As we mentioned above, Figure 3 in fact shows that prompts that have low training error can actually have high test error (e.g., negative correlation) whereas PAC-Bayes bounds correlate well with generalization.
>
> > Does prompting the language model differently induce a better prior?
>
> This is an excellent question! We have tried prompting with “this is an image of '' following similar reasoning that you provided, but did not improve the performance of the method in our initial experiments. We are currently running some more ablation and hope to finish before the discussion period ends.
>
>
> We hope these responses address your concerns and clarify our approach. Thanks again for your review and we are open to further discussion.

---

> > ### Author Response · Authors · 2023-11-21
> > **Revision uploaded**
> >
> > We have uploaded the revised version (all revisions are in orange). In particular, we have changed the caption of figure 3 to make things clearer. We hope that the revision addresses your concerns and would be happy to make further revision. We look forward to hearing from you. Thank you!

---

> ### Author Response · Authors · 2023-11-22
> **Additional experiments about random label and prompting language model to induce a better prior**
>
> We have uploaded a revision to include two sets of experiments.
>
> 1. We ran linear probing on cifar10 with random labels and found that it was only able to achieve 13.6% accuracy. This supports the claim that Greedy is not too simple of a search approach to fit the random labels as other more complex methods also cannot, since even linear probing can't find a good classifier. This is added to the **Fitting random labels** part of Section 5.
>
> 2. We also prompted the language model with 10 different prompts (e.g., `a photo of a [...]` and `a blurry photo of a [...]`) and showed the results in Appendix B.2. We observed that the prompting does not significantly affect the performance of the final prompts.
>
> Please let us know if they address your concerns. Thank you!

---

> > ### Comment · Reviewer_q9H1 · 2023-11-22
> > **update**
> >
> > Thanks to the authors for addressing my concerns. I've increased my score.

---

### Official Review · Reviewer_bDCR · 2023-11-01

**Soundness:** 3 good
**Presentation:** 3 good
**Contribution:** 3 good
**Rating:** 8
**Confidence:** 2

**Summary:**

This paper proposes a new method to compute the generalization bound: by utilizing a language model to construct the prior and posterior distribution in the PAC-Bayes bound, the authors get a better generalization bound for prompt engineering for Vision-language models.

**Strengths:**

The problem is an important problem: understanding LLMs and other big models is an important problem. Computing or adding more insight into the generalization of these models is also important. The methodology seems interesting to me. The presentation is good.

**Weaknesses:**

It seems that the numbers between the UC and the PAC-Bayes, and I feel like there is not too much difference (most of them are within a factor of 2 or 3). I don't think the new generalization bound result is significantly different (although the method seems interesting to me), since scaling up the data by 4 to 9 times can reach nearly the same generalization error. Maybe more results with more scales helps.

**Questions:**

Please see the weakness section, which criteria should be considered to test that a new generalization bound is significantly better? Also, is it possible to use similar ideas to analyze the (hard) prompt tuning in natural language processing? Besides, using the pre-trained LLaMA can improve the PAC-Bayes bound, is it some form of transferring the ''generalization problem'' of the prompt engineering to the generalization problem of the pre-trained language model?

---

> ### Author Response · Authors · 2023-11-17
> **Response to Reviewer bDCR**
>
> Thank you for your insightful feedback on our paper! We are glad that you find the problem important and our methodology interesting. We will try to address your concerns as follows:
>
> > scaling up the data by 4 to 9 times can reach nearly the same generalization error
>
> The goal of deriving generalization bounds is to study the finite-sample behavior of a method, since in the limit of infinite data, all (reasonable) complexity terms would vanish. Besides, increasing dataset size significantly is not always practical due to resource constraints In many scenarios.
>
> > which criteria should be considered to test that a new generalization bound is significantly better?
>
> This is a great question. The effectiveness of a generalization bound is typically assessed by its tightness (how small it is given a finite number of data) and its ability to inform model selection. In our case, both UC and PAC-Bayes bounds over the discrete tokens are significantly tighter than the bounds reported in prior works, and our PAC-Bayes bounds are shown to correlate well with the test performance (Figure 3). This tightness and correlation with test performance underline the bounds' effectiveness and novelty.
>
> > is it possible to use similar ideas to analyze the (hard) prompt tuning in natural language processing?
>
> This is a very interesting prospect, but the majority of existing work on generalization bounds is done on image classification so it’s unclear, at least not immediately obvious, how to apply them properly to NLP tasks. To keep the scope of the paper focused, we feel that this is better suited for future work.
>
> > Besides, using the pre-trained LLaMA can improve the PAC-Bayes bound, is it some form of transferring the…
>
> We are not sure if we understand this question. We view the fact that prompts with high likelihood also tend to give better bounds as evidence that the more a prompt looks like "natural language" the more likely it is to generalize well since CLIP is trained in natural language and LLaMA is trained to model natural language. So maybe the answer is in some sense it is " some form of transferring the 'generalization problem' ", but at a very abstract level.
>
> We hope these clarifications address your concerns and would be happy to answer further questions. Thank you again for your review!

---

> > ### Comment · Reviewer_bDCR · 2023-11-22
> > **Reply**
> >
> > Thanks for your reply. I think my concern is partially addressed. After thinking for some time, I will increase my score to 8.

---

### Official Review · Reviewer_DH9x · 2023-11-05

**Soundness:** 3 good
**Presentation:** 3 good
**Contribution:** 3 good
**Rating:** 8
**Confidence:** 3

**Summary:**

The paper constructs (PAC-Bayes) generalization bounds using priors constructed from joint vision-language models. The found bounds are tighter than common bounds used for image classification with neural networks from scratch, which are usually tight unless data-dependent.

**Strengths:**

- interesting and alternative perspective on prompting and construction of PAC-Bayesian bounds for modern deep learning methods
- theoretical development well-written and easy to follow
- find that prompt engineering for image classification provides relatively strong generalization guarantees and almost no overfitting in practice

**Weaknesses:**

- introduction/abstract somewhat oversell the paper and contradict later factual statements made in the paper (see issues of presentation below)
- linear probe significantly better than using the greedy prompt search as shown in Tab. 3, which calls the performance of the greedy method into question and makes it unclear whether one should even use prompt engineering in such cases then despite the tighter bounds

#### Issues of presentation
- introduction & abstract dwell on the issue of classical deep learning PAC Bayes bounds, which are usually vacuous, unless conditioned on data. However, pretraining on a massive data set is rather similar to conditioning on data so I find this a bit inconsistent. I suppose it would be good to define data-dependent clearly if it applies to standard bounds but not the proposed one. Especially considering the likely case of data leakage into clip embeddings, claiming that the proposed method is not data-dependent seems inappropriate.
- title claims "understanding" but the paper did not really provide a new understanding of prompt engineering, I would argue, and the authors seem to agree with this as can be seen in the 2nd paragraph of the conclusion. The 3rd paragraph of the conclusion is much more clear and honest given the results.

**Questions:**

I am mostly confused about the claims made in title, abstract, and introduction and otherwise found the paper interesting. The two things that bother me are the question of "data-dependence" and "understanding prompt engineering".
1. data-dependent PAC-Bayes bounds would learn a prior on parameters based on a small pretraining dataset. This is extremely similar to the approach of pretraining a vision-language model. Would you disagree with this or see this differently?
2. I did not have the feeling to improve my understanding of prompt engineering from the paper but rather has brought forward many interesting other aspects, especially that we can construct significantly tighter bounds for image classification for prompt-based classification than training from scratch.

---

> ### Author Response · Authors · 2023-11-17
> **Response to Reviewer DH9x**
>
> Thank you for your thoughtful review! We are glad that you find the perspective of this work interesting. We will try to address your concerns below.
>
> > linear probe significantly better than using the greedy prompt search as shown in Tab. 3
>
> The goal of the paper is not to argue the superiority of prompt engineering over finetuning. The experiment's goal is to highlight that prompt engineering has a much smaller hypothesis space than even linear probing so they naturally generalize well. In most applications of pretrained models, finetuning with gradients would generally outperform prompting, but prompting has the benefit of being extremely simple and intuitive (i.e., requiring only language-based descriptions of classes), and can work even in the absence of an explicit finetuning set or parameter update (though such a finetuning set would, of course, be needed to compute our actual bounds).  Indeed, we feel it is largely for these reasons that prompt engineering is such a dominant approach in practice.  Further, although we highlight prompt search as a method (which again would require a finetuning set), the most common usage of prompt engineering is usually in this more manual setting.
>
> > introduction & abstract dwell on the issue of classical deep learning PAC Bayes bounds…
>
> “Data dependent” generally refers to the fact that the prior is dependent on the data from the task of interest (e.g., CIFAR10) [1]. Concretely, one would partition the training set into two disjoint sets and use one set to define the prior and the other set to learn the model. This definition does not preclude using other data (unless of course there is data leakage). A recent work [2] showed that CLIP still works well on Imagenet even if one removes instances similar to ImageNet, so we believe that data leakage may not be a significant issue for CLIP’s performance. We will revise the text to make it clearer what we mean by data-dependent (currently still revising).
>
> > title claims "understanding" but the paper did not really provide a new understanding of prompt engineering…
>
> The core insight we hope to highlight is that prompt engineering operates in an extremely “small” hypothesis space by the standard of deep learning, which comes with good generalization properties out of the box. We believe that this is an interesting, if not valuable, observation that is not well-known currently in the community. The goal of the conclusion is to highlight that understanding prompt engineering requires understanding something else beyond generalization which is in line with our title.
>
> > This is extremely similar to the approach of pretraining a vision-language model. Would you disagree with this or see this differently?
>
> This is a great question. They are similar in the sense they both leverage some data that are not the training data but there are some subtle differences. In the data-dependent PAC-Bayes literature, this pre-training data is chosen to be from the same distribution as the actual data (e.g., partitioning the training data and using the small subset to learn the prior). This ensures that the bound is tight because the weights would not move “far” from the prior since the two partitions are from the same distribution.
> This is generally not the case for VLM models where it is trained on a lot of data that are not from the same distribution and not with the same objective. This difference may seem subtle but it is crucial in practice and in theory since we have a large amount of data from different tasks but not necessarily from the same task.
>
> > I did not have the feeling to improve my understanding of prompt engineering from the paper but rather has brought forward many interesting other aspects…
>
> The understanding we hope to put forth is the fact that prompt is a naturally good hypothesis space for generalization because it is very “small”. The tight generalization bound is a testament to how small the hypothesis space is compared to the conventional parametric models. We believe that this claim is substantiated by our experiments.
>
>
> We hope these clarifications address your concerns and would be happy to answer further questions. Thank you again for your review!
>
> **reference**
>
> [1] On the role of data in PAC-Bayes bounds. Dziugaite et al.
>
> [2] Does CLIP's Generalization Performance Mainly Stem from High Train-Test Similarity? Mayilvahanan et al.

---

> > ### Author Response · Authors · 2023-11-21
> > **Revision uploaded**
> >
> > We have uploaded the revised version (all revisions are in orange). In particular, we have revised the introduction to clarify what we mean by data-dependent. We hope that the revision addresses your concerns and would be happy to make further revision. We look forward to hearing from you. Thank you!

---

### Comment · Area_Chair_QX8z · 2023-11-10
**Authors-Reviewers discussion starts today, ends on Nov 22**

Dear authors and reviewers,

@Authors: please make sure you make the most of this phase, as you have the opportunity to clarify any misunderstanding from reviewers on your work. Please write rebuttals to reviews where appropriate, and the earlier the better as the current phase ends on Nov 22, so you might want to leave a few days to reviewers to acknowledge your rebuttal. After this date, you will no longer be able to engage with reviewers. I will lead a discussion with reviewers to reach a consensus decision and make a recommendation for your submission.

@Reviewers: please make sure you read other reviews, and the authors' rebuttals when they write one. Please update your reviews where appropriate, and explain so to authors if you decide to change your score (positively or negatively). Please do your best to engage with authors during this critical phase of the reviewing process.

This phase ends on November 22nd.

Your AC

---

### Meta-Review · Area_Chair_QX8z · 2023-12-05

**Metareview:**

This meta-review is a reflection of the reviews, rebuttals, discussions with reviewers and/or authors, and calibration with my senior area chair. This paper constructs PAC-Bayes generalisation bounds using priors constructed from joint vision-language models. There is a consensus among reviewers to  praise this solid work, on an increasingly popular topic, with interesting theoretical contributions although holding under stringent assumptions which might not be valid in practical cases.

**Justification For Why Not Higher Score:**

Solid work, on an important topic, with interesting theoretical contributions although holding under stringent assumptions.

**Justification For Why Not Lower Score:**

Solid work, on an important topic, with interesting theoretical contributions although holding under stringent assumptions.

---

### Decision · Program_Chairs · 2024-01-16

Accept (poster)